# Lighting Professionals versus Light Pollution Experts? Investigating Views on an Emerging Environmental Concern

**Nona Schulte-Römer** [1,*] **, Josiane Meier** [2] **, Etta Dannemann** [3] **and Max Söding** [1,2]

1 Department of Urban and Environmental Sociology, Helmholtz Centre for Environmental Research, 04318 Leipzig, Germany; m.soeding@campus.tu-berlin.de
2 School of Planning-Building-Environment, Technische Universität Berlin, 10623 Berlin, Germany; josiane.meier@tu-berlin.de
3 Studio Dannemann, Baerwaldstraße 63A, 10961 Berlin, Germany; etta@studiodannemann.de
* Correspondence: nona.schulte-roemer@ufz.de

**Abstract:** Concerns about the potential negative effects of artificial light at night on humans, flora and fauna, were originally raised by astronomers and environmentalists. Yet, we observe a growing interest in what is called light pollution among the general public and in the lighting field. Although lighting professionals are often critical of calling light 'pollution', they increasingly acknowledge the problem and are beginning to act accordingly. Are those who illuminate joining forces with those who take a critical stance towards artificial light at night? We explore this question in more detail based on the results of a non-representative worldwide expert survey. In our analysis, we distinguish between "lighting professionals" with occupational backgrounds linked to lighting design and the lighting industry, and "light pollution experts" with mostly astronomy- and environment-related professional backgrounds, and explore their opposing and shared views vis-à-vis issues of light pollution. Our analysis reveals that despite seemingly conflicting interests, lighting professionals and light pollution experts largely agree on the problem definition and problem-solving approaches. However, we see diverging views regarding potential obstacles to light pollution mitigation and associated governance challenges.

**Keywords:** light pollution; sustainable lighting; light planning; expert survey; ALAN

## 1. Introduction

Light pollution broadly describes unwanted or excess artificial lighting at night, and the negative effects artificial illumination can have on humans and the living environment. While the concept is rather ill-defined, it has received increased public attention in recent years. The concerns are reflected in growing numbers of media reports, fuelled by public campaigns and findings based on scientific evidence from various disciplines. Biologists have highlighted the negative effects of artificial light at night on species as diverse as birds, bats, fish, insects, water organisms, mammals and plants [1–3]. Medical research suggests that light at the wrong time confuses the human circadian rhythm with negative effects on people's sleep, which may impact their health [4,5]. Astronomers highlight the reduced visibility of the night sky [6], and in the social sciences and humanities, natural darkness is being rediscovered and re-evaluated as a cultural asset and distinct social space [7,8]. These multifaceted issues reverberate in civic complaints about light nuisances in urban and natural environments, and in new policies for outdoor lighting such as the national light pollution laws in France and Slovenia [9,10]. Together with concerned individuals and advocacy groups, researchers who take a critical view of artificial light at night can be considered as an emerging community of

light pollution experts. They draw attention to the unwanted side effects of artificial illumination by producing, exchanging and publicizing information and knowledge via social media, mass media and scientific journals, and at events. They also actively propose new planning and policy approaches as they question established light practices and reasons for illuminating public spaces, buildings, signs or landscapes.

The new notion that light is also a pollutant problematizes artificial lighting, which is usually overwhelmingly positively connoted [11–13]. It is therefore not surprising that lighting professionals, who develop lighting technology, sell lighting products, and plan and design lighting schemes, have not been the loudest voices in debates about light pollution. Nevertheless, lighting designers, light planners and manufacturers, who are traditionally concerned with the improvement and dissemination of light sources and installations, have begun discussing scientific evidence for environmental and health concerns in conferences and professional journals, and are beginning to adjust their practices, products and professional education accordingly [14,15].

The recognition of the problem by those who illuminate and create lighting is highly relevant when it comes to tackling the issue of light pollution. However, this raises the question of how the views of actors in the lighting field compare to those of the researchers and activists that have adopted a critical stance toward lighting. How do the professional interests of lighting designers, planners and manufacturers align with the recommendations and claims of light pollution experts? Where do they agree or disagree? What are the practical and political implications of their respective perspectives?

In this paper, we explore and contrast the views of lighting professionals and light pollution experts with the goal of highlighting common ground and conflicting views. Our analysis is grounded in qualitative research and professional experience, and draws on the results of an online expert survey on light pollution. Conducted in 2018, it was completed by 205 participants. They include lighting designers, planners and lighting engineers or manufacturers, which we categorize as "lighting professionals" (n = 67), and respondents who work on light pollution issues and largely have astronomical and environmental backgrounds, which we identify as "light pollution experts" (n = 89). Our findings suggest that lighting professionals surprisingly often agree with light pollution experts, not only in their views regarding light pollution, but also when it comes to recommending solutions to the problem. Their views diverge more when it comes to identifying obstacles to light pollution mitigation. These results also have practical relevance, as they reveal which policy options for sustainable lighting can find support in both groups and where alternative or opposing views should be tested and further discussed.

## 2. Materials and Methods

This paper is part of a larger research project, and is informed by our previous professional and research experience in the fields of lighting and light pollution [16]. The idea for this study emerged from our observation that the lighting community and the emerging community of light pollution experts engage in arenas that are in many ways worlds apart, but at the same time, closely connected through their focus on artificial light. The outsets of the two groups seemingly contradict each other: while lighting professionals earn their living creating light, light pollution experts are concerned with reducing artificial light at night. At the same time, light pollution has clearly become a point of debate in the lighting world, and the light pollution community aims to include lighting professionals [17,18]. Based on these observations and our empirical and practical knowledge of lighting practices and debates around light pollution and its mitigation, we developed a set of theses in order to explore how the views and goals of lighting professionals and light pollution experts compare. These assumptions were then tested in our online expert survey.

### 2.1. Data Collection Based on an Online Expert Survey

The questionnaire was developed for "experts", i.e., respondents with practical or theoretical knowledge of and interest in artificial lighting and/or light pollution [19]. The survey design ensured

this expertise in three ways: First, respondents were asked to outline their "light-related activities". Second, some questions were highly specific and demanded an in-depth understanding of lighting issues, as pre-testers confirmed. Third, we consciously chose to use the term "light pollution", including in the survey's title. By using the term so explicitly, we specifically addressed respondents who are familiar with the issue. The survey was only distributed in English, which could possibly result in an under-representation of experts that are not part of the relevant English-language discourse.

The expert survey was launched in March, 2018, and was online for two months. The invitation was circulated internationally via e-mail, twitter and professional networks, creating a snowball effect (more information at [16]). Clearly, this sampling strategy could not produce a representative sample. However, in line with our exploratory approach, it allowed as many experts as possible who wished to share their opinions on light pollution to do so.

The questionnaire contained both quantitative and qualitative elements. Participants were asked to tick boxes to describe their personal background and to evaluate specific aspects around the issue of light pollution on Likert scales from one to five. In addition to single and multiple choice questions, open questions allowed the respondents to answer using their own words and to add aspects not included in our suggested answers for closed questions. Our questions covered three thematic areas: (1) The definition of and opinions on light pollution; (2) the governance challenge in terms of main obstacles, clashing interests and responsibilities; (3) possible solutions in the form of recommendations.

## 2.2. Group-Specific Data Analysis

The survey was completed by 205 participants. For the stakeholder-specific analysis, we identified and created the group categories "lighting professionals" and "light pollution experts" within our sample. While the concept of "lighting professionals" is quite straightforward and includes people who professionally plan, design, or produce artificial light and lighting technology, the notion of "light pollution experts" calls for an explanation. We conceptualize this group as a heterogeneous "issue public" [20] consisting mostly of astronomers, conservationists, natural and social scientists who problematize artificial light at night (ALAN) from their various viewpoints. In reality, the two groups can overlap. At an individual level, there are lighting professionals that engage heavily in raising awareness for light pollution and developing solutions for its mitigation, as well as persons with backgrounds in fields such as astrophysics or biology who have acquired detailed knowledge of lighting technology and lighting practices and e.g. advise municipalities on sustainable lighting. While we are aware of these overlaps, we nevertheless distinguish between lighting professionals and light pollution experts on the basis of their different foci and fields of activity. Table 1 outlines the answers to both closed and open questions, on the basis of which we categorized the respondents.

The categorization process left us with a sample of 156 respondents: 89 light pollution experts and 67 lighting professionals. The respondents were aged between 20 and 79, and about one third was female. Most of them (101; 65%) were based in Europe, 29 (19%) in Anglo America, 13 (8%) in Australia/New Zealand, 5 (3%) in Middle Eastern or African countries, 4 (3%) in Latin America and 3 (2%) in Asia (1 answer missing).

Based on this data, we performed our analysis in three steps using the software R. First, we studied relative frequencies and mean values to identify answers where the two groups' views converge or diverge. Second, where mean values and relative frequencies differed considerably, we performed regression analyses to test the impact of participants' occupations and whether divergences can be better explained by other independent variables. To be more precise, we tested in binomial logistic regression models for the impact of occupation (light pollution experts or lighting professionals), place of residence (Europe or Anglo America/Oceania and Anglo America or Europe/Oceania as well as urban or not), age (in years), first encounter with light pollution (number of years) and gender (male or female, the two "other" responses were considered as "missing", see Table S1 and Figure S1).

We found that occupational backgrounds were indeed the best predictor, while age, gender or place of residence were rarely significant. The dependent variable was the approval of the respective

item (4 and 5 on a scale of 1 to 5). Third, open statements helped us confirm shared opinions and understand differences. In line with our research interest and in light of our non-representative sample, we focused more on converging views than on differences, which we had expected to be dominant, as outlined in the following. Figures are produced with the Microsoft software Excel.

**Table 1.** Categorization of respondents based on "light-related" and "other" main occupations.

| | Categories Based On Closed Questions Regarding The Respondents' Light-Related Main Occupation | Categories Based On Open Answers Regarding "Other" Than Light-Related Main Occupations. |
|---|---|---|
| Lighting professionals (N = 67; 25 females, 41 males, 1 other; aged between 26 and 79) | • Architectural and decorative lighting design (indoor/outdoor)<br>• Functional light planning (streets, parking lots, etc.)<br>• Development of urban lighting concepts/master plans<br>• Light art/artistic work using light (no answer)<br>• Marketing and/or the sale of lighting products<br>• Lighting technology research and development | • Providing of lighting or information on lighting (via online platforms, electronics engineering services, as part of energy provision and consulting in developing countries). |
| Light pollution experts (N = 89; 23 females, 65 males, 1 other; aged between 20 and 75) | • Environmental protection related to lighting<br>• Raising awareness for light pollution | • Astrosciences and -technology related occupations (e.g., professional or amateur astronomers, airglow researchers, educators in planetariums)<br>• Environment-related occupations (including scientific work in biology, chronobiology or the environmental sciences, educational work in nature reserves and parks, journalism, etc.)<br>• Other research related to the effects of lighting (university lecturers and researchers of various disciplines, including law, archaeology, history, sociology, physiology, etc.)<br>• Raising awareness for light pollution (non-profit activists, voluntary dark-sky educational work, etc.)<br>• Retired respondents with an interest in astronomy and light pollution mitigation. |

## 3. A Conceptual Framework: Exploring Expert Perspectives on Light Pollution

In recent years, initial studies have explored the general populations' views on lighting and light pollution. For instance, Lyytimäki and Rinne [21] carried out an online survey to understand how people in Finland perceive and respond to light trespass and other light nuisances (n=2053). In Germany, Besecke and Hänsch [22] explored how residents of an inner-city street of Berlin and inhabitants of a nearby suburban community perceived light and darkness before and after street lighting refurbishments to LED lighting. Green et al. [23] used ethnographic data, household survey and documentary sources to explore responses to street lighting reductions in eight areas of England and Wales. This study complements this strand of research by providing results on *expert* perspectives on the topic. Expert perspectives are relevant as outdoor lighting has long been delegated to expert systems and is only just re-emerging as a public issue [24]. In contrast to studies of the general public, which are methodologically challenging as they demand asking people about their implicit practical knowledge about lighting [25], focusing on experts makes it possible to investigate the issue—including its technical aspects—in more depth and detail, given the respondents' higher level of previous engagement with specificities of the topic. Other than most laypersons, they pay attention

to light and darkness and also have a vocabulary to express their observations and feelings about lighting. Moreover, expert opinions are also particularly relevant as they shape realities of artificial light and natural darkness by planning, designing or contesting lighting.

Since we could not draw on existing expert surveys, we had to come up with our own conceptual framework for assessing the group-specific views. Social-scientific theory suggests that expert groups form "communities of practice" with specific understandings and shared views on their respective issues of concern [26]. Recent discourses in the two stakeholder groups allowed us to develop the three thematic areas covered in the survey based on explicit assumptions, as outlined below (Table 2). In line with our empirical observation, Challéat and colleagues [19] have described two camps vis-à-vis lighting in France: lighting professionals who promote a technical view on "light nuisance" and astronomers, conservationists and citizens who take an environmental stance against "light pollution".

### 3.1. Light Pollution Experts and the Negative Side-Effects of Artificial Light at Night (ALAN)

To conceptualize the views of the light pollution experts on a global scale, we can draw on the growing body of scientific literature on the effects of artificial light at night (ALAN), which is also the basis for social scientific and planning discourses on ALAN as well as for activists. This interdisciplinary and emerging field can be roughly divided into three areas: Research mainly by astronomers and astrophysicists on sky glow and light trespass as an impediment to the observation of the universe; biological research investigating the impact of ALAN on individual animal and plant species, and increasingly, on ecosystems; medical research exploring the chronobiological hormonal effects that are triggered by ALAN and are suspected to increase the risk of depression, cancer, cardio-vascular diseases and obesity.

Experts who work on these issues have significantly shaped the notion of light pollution. Astronomers, both professionals and amateurs, are a driving force behind initiatives for dark-sky protection. With the spread of electric lighting in the early 20th century, they were among the first to criticize and quantify the reduced visibility of celestial objects [27]. Today, they explore and develop new instruments and methods for assessing the illumination of the night sky [6,28–30]. They also warn that blue-rich LED light scatters more strongly in the atmosphere and will, in combination with rebound effects, increase not only sky glow but also glare [31,32].

Biologists and ecologists have been studying the effects of artificial light at night on birds, insects, aquatic organisms, reptiles, mammals and plants to understand and assess its impact on these different species as well as entire ecosystems. In recent years, they have substantiated their suspicion that light affects animal behavior (e.g., through distraction) and disturbs the circadian rhythm of living organisms more generally, both with negative consequences for the finely orchestrated processes of all life that have evolved over millennia under planetary rhythms of light and darkness [1]. All light spectra can be potentially harmful, as different species are sensitive to different types of light. Therefore, full-spectrum light sources and blue-rich light seem to be more problematic than light with a narrow spectrum and longer wavelengths, as these will probably affect more species [33,34]. Since circadian processes also govern the human body, exposure to ALAN, and particularly to blue-rich light, has also become a public health concern [5]. Medical studies suggest that ALAN is a stressor for people who work night shifts or are exposed to blue-rich light at night, such as that emitted by LED lighting [35].

Although scientific evidence on the biological impacts of ALAN is still patchy, many biologists and physicians have come to take a precautionary stance and promote the protection or restoration of natural darkness or reduced light levels. In that and in their reservations regarding blue-rich lighting, they share views and goals with astronomers, as well as with actors that engage critically with the illumination of the night from other viewpoints, such as culture or aesthetics [3,36]. In the latter respect, it is frequently highlighted that we are losing the experience of natural darkness and the visibility of the stars and planets, which has been a key to human civilization [37].

Light pollution experts also actively recommend, develop and test counter-measures. They develop models to assess the scope and effects of the problem, as well as the viability of solutions [3,27,32,38]. They criticize the fact that existing lighting technology, lighting standards and regulations are not sufficient and that they should be updated to acknowledge issues of light pollution [39]. Advocacy organizations such as the International Dark-Sky Association (IDA), but also researchers in the ALAN community, address the wide-spread ignorance of the issue and actively engage in raising awareness for light pollution (e.g., ida.org, cost-lonne.eu, stars4all.eu). Finally, light pollution experts are actively involved in shaping lighting technology (e.g., shielded luminaires, PC amber LEDs) and the governance of lighting via tools that range from technical recommendations (e.g., avoidance of light above the horizontal and blue-rich lighting) to education (e.g., in observatories) and mandatory legislation [40–42].

**Table 2.** Overview of our empirically grounded assumptions regarding group-specific views.

| Assumptions | Light Pollution Experts | Lighting Professionals |
|---|---|---|
| **Basic assumptions** | | |
| What are the group-specific interests? | • Reduce artificial light at night, stop loss of the night.<br>• Acknowledge and tackle the problem in projects, guide-lines, rules and regulations. | • Sell lighting expertise in design and building projects.<br>• Promote good, visually comfortable, aesthetically appealing light and darkness. |
| **Problem perception** | | |
| What is light pollution? | • All artificial light at night is a form of pollution. | • It depends on the situation, whether light is pollution.<br>• Given the many positive effects of lighting, the term 'pollution' is inappropriate. |
| Why is it a problem? | • ALAN can have negative effects on flora, fauna, humans and ecosystems.<br>• Experience of natural darkness and the visibility of night-time skies are lost. | • Visual discomfort (e.g., due to glare) and light trespass.<br>• Negative effects on people's sleep.<br>• Unnecessary energy consumption and cost. |
| **Governance challenge** | | |
| What are the obstacles? | • Lack of awareness and knowledge amongst decision-makers, lighting professionals and light users.<br>• Worldwide increase in blue-rich white LED lighting which intensifies the problem. | • Lighting professionals have solutions, but they are not invited.<br>• Adequate technology and best practices are available, but need to be disseminated. |
| **Possible solutions** | | |
| Who is responsible? | • Actors in the lighting field, regulators and light users. | • Actors in the lighting field. |
| What should be done? | • Reduce artificial light at night, avoid blue-rich light.<br>• Develop policies for mitigating light pollution, including hard regulations.<br>• Develop better technology that reflects the state of knowledge regarding the negative effects of lighting. | • Encourage sustainable instead of cheap solutions.<br>• Apply existing knowledge and recommendations.<br>• Plan and design according to the state of the art in lighting.<br>• Use smart technology and apply adaptive lighting. |

Based on these discourses and developments, we assumed the following:

1.  Regarding the definition of light pollution, we expect that light pollution experts contend that *all artificial light at night is pollution*, because even small amounts of ALAN are an alteration of natural darkness and may affect living beings and the possibility to observe the night sky. In terms of the problem's dimensions, we assume that they highlight potential non-visual effects of light on flora, fauna, humans and ecosystems, as well as the cultural loss of natural darkness and star-filled skies.
2.  Regarding the governance challenge, we expect that light pollution experts call for more political commitment and highlight the need to raise awareness for light pollution, to provide more guidance and information to decision makers.
3.  In terms of the problem's solutions, there seems to be a widely-shared consensus in the light pollution community that *systemic change* is necessary. We therefore assume that light pollution experts recommend more sustainable technology, better education and information, as well as better technical guidance, lighting standards and stricter regulations.

### 3.2. Lighting Professionals and the Art of Planning, Designing and Manufacturing Light

Lighting professionals' perspectives are less obvious, as they often do not refer to light pollution when they write about potential negative side-effects of lighting. As a lighting designer remarked in response to our survey invitation, "the term 'light pollution' is an evocative phrase for many lighting designers, including me. Our stance is that light is a pure and natural phenomenon, and the 'pollution' angle comes from the misuse of light, or light in the wrong place. We feel that the terms 'obtrusive light' and/or 'light trespass' are more fitting." In a commentary published in Nature, lighting designer Zielinska-Dabkowska outlines the potentially negative health effects of lighting without mentioning the term "light pollution" even once [43].

Looking at lighting practices and projects, energy and cost efficiency constitute long-standing benchmarks that can be linked to light reduction. Accordingly, the British Institution of Lighting Professionals (ILP) argues in its Guidance Notes for the Reduction of Obtrusive Light: "Do not 'over' light. This is a major cause of obtrusive light and is a waste of energy. There are published standards for most lighting tasks, adherence to which will help minimize upward reflected light." [44] (p. 1111).

This recommendation also reflects the basic stance that 'good' lighting means providing appropriate lighting for a given time, place and task. Light engineering and illuminating societies develop technical standards to provide orientation towards achieving this complex goal (ies.org, theilp.org.uk and licht.de). Lighting design associations and expert networks provide information, education and exchange platforms that enable their members to plan and design light in situation-specific ways and according to their clients' needs (e.g., iald.org, pld-c.com, luciassociation.org). Light manufacturers also subscribe to this goal which allows them to further develop and diversify their product lines, for instance with a focus on heath or enhanced work performance [45,46].

Knowing how to accomplish 'good' lighting is considered a characteristic and distinctive skill of lighting professionals, which qualifies them more than electricians, civil engineers, architects or private home owners to illuminate the world at night. However, in reality, such explicit lighting expertise is often ignored or only invited in the final stages of building or design projects. Therefore, light planners, specifiers and lighting designers often describe light pollution as a problem of missed opportunities: Short-sighted cost-benefit calculations, lack of expertise and time pressure lead to suboptimal solutions that cause nuisances and unwanted side effects.

Professional experience and knowhow appear to be particularly relevant in light of two major developments: For one, climate change policies affect lighting practices in the form of economic incentives, but also product bans like the out-phasing of the incandescent light bulb [25]. For another, light-emitting diodes (LEDs) constitute a disruptive technological innovation [47]. LED technology is widely seen as an energy efficient means to provide "the right light at the right place at the right

time", since LEDs are highly directional and can be digitally controlled and adapted in brightness and color temperature. They thus open new business opportunities, which also relate to issues of light pollution. For instance, light manufacturers work on optical systems that reduce glare, conduct their own research on the non-visual effects of blue-rich LED light and offer new products, including PC amber LEDs, to meet the demands of dark-sky friendly lighting schemes.

Finally, new conceptual approaches to lighting are relevant to light pollution debates. First, LED lighting is promoted with visions of adaptive "smart" lighting that responds to lighting needs, thereby reducing excess light [47,48]. The notion of "human-centric lighting" highlights the relationship between light and well-being, thereby widening the thus-far dominant focus on more functional aspects like visibility and safety [49]. In this concept, lighting professionals show a preference for adaptable white light sources with a continuous spectrum, which provide better color rendering than the widely-used sodium vapor street lamps, and are thus assumed to enhance visual comfort in outdoor spaces. Last but not least, lighting designers also highlight the value of darkness, but more for aesthetic than for environmental reasons, which are less prominently voiced in lighting projects [13] (pp. 182–187).

Based on these observations, we assumed the following:

1. Regarding the problem's definition, we expect lighting professionals to argue that *it depends on the situation* whether light is pollution, or even to *reject the notion that light can be pollution* altogether. In terms of the problem's dimensions, they will probably be more concerned with reducing energy consumption and improving humans' visual comfort (full light spectrum, no glare) and well-being than with protecting natural darkness and star-filled skies or reducing potential negative effects on flora and fauna.
2. With regard to the governance challenge, we expect that lighting professionals take responsibility and make the mitigation of light pollution their own task, as it calls for professional skills and constitutes a potential business case. It seems likely that they will argue that light pollution would not be a problem if lighting were properly planned and designed by experts.
3. Regarding the problem's solutions, we accordingly assume that lighting professionals blame procedural and project-related shortcomings like the lack of lighting expertise in building projects and call for an earlier and more consistent involvement of professionals. We further expect them to rely on the self-regulatory functions of their professional institutions and the state of the art in their professional domain (existing guidelines, best practices, innovative technological solutions and products) rather than calling for 'external' intervention via stricter rules and more regulation.

## 4. Findings: Where Experts (Dis)agree

Overall, our results show that light pollution experts generally express stronger opinions on the issue than lighting professionals. We further see that views are more consensual regarding understandings of the problem and possible solutions than they are regarding the question of why it is difficult to tackle light pollution, that is, the governance challenge. To better understand the contexts in which the survey respondents form their opinions, we asked them to specify how their professional or voluntary light-related work is affected by current trends (Figure S2 and Table S2). The results show that the two groups perform their activities under the impression of relatively similar dynamics, especially the introduction of LED lighting in outdoor spaces, irrespective of their geographical background.

### 4.1. What is Light Pollution?

To test our assumptions regarding problem perceptions, we asked the respondents to express their opinions about the notion of light pollution. Not surprisingly, light pollution experts were more critical of ALAN than lighting professionals (Figure 1): 47% of them consider *all outdoor lighting as pollution*, while 53% think that *it depends on the situation*. Conversely, given the frequently encountered

scepticism towards the term, we were surprised that 28% of the lighting professionals in our sample even subscribed to the absolute view that "all outdoor lighting after dark is a form of pollution." Of the respondents, 66% think it depends on the situation, and 3% of lighting professionals answered that "outdoor lighting is never pollution".

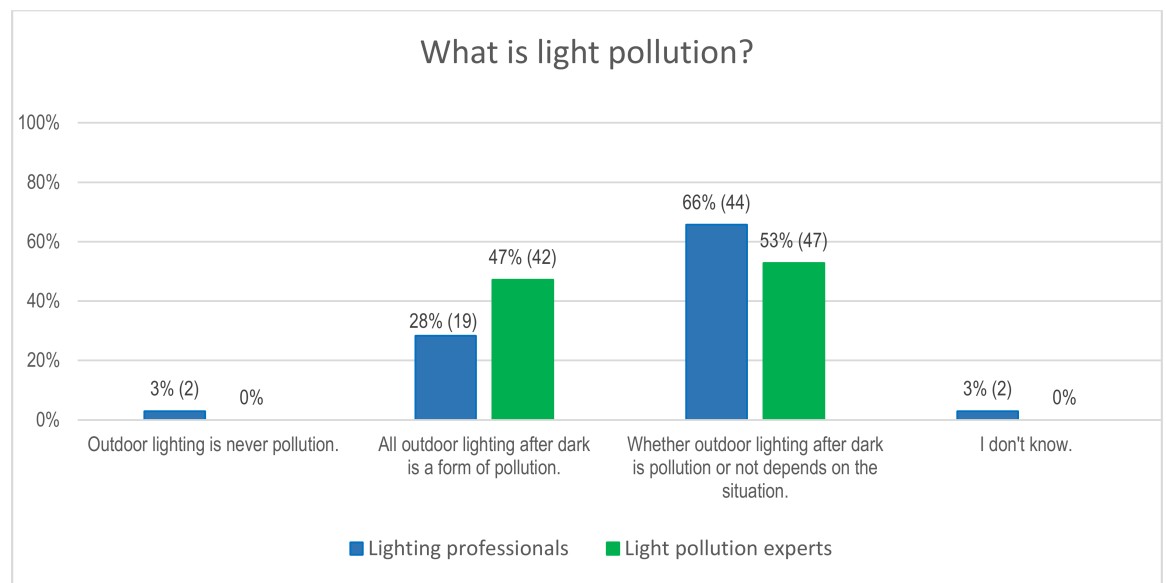

**Figure 1.** Problem definition: "What is your personal opinion regarding light pollution?" Answers in percentage per group and (absolute numbers).

In a follow-up question, we then asked those who had answered "it depends" to specify problematic situations (multiple choice). Figure 2 shows that light pollution experts find more situations problematic: 66% of them ticked nine out of eleven possible answers, while only 23% of the lighting professionals agreed with nine answer options. The majority of lighting professionals agree with light pollution experts, albeit to a lesser extent, that *light is pollution when it enters areas where it is unwanted* (light trespass), is *not used*, *obscures the visibility of the stars* and *produces glare*.

Discrepancies between the groups vary. They are smallest when light is not automatically considered pollution, such as *colorful lighting* or *moving and blinking lights*. Opinions differ most when it comes to the illumination of specific spaces like *natural areas* or *close to bodies of water* or *observatories* (inter-group differences of more than 45 percentage points) and lighting-technological aspects like *blue-rich lighting, color temperature and glare* (inter-group differences between 37 and 44 percentage points).

The discrepancy allows two interpretations: Firstly, the comparatively low recognition of problematic spaces among lighting professionals might be a sign of their unawareness or indifference with regard to effects of ALAN on water organisms, flora and fauna in general, or astronomical observations. However, as we will see below, answers to follow-up questions do not support this interpretation. Secondly, we might conclude that light pollution experts lean towards more definite essentialist understandings of the problem, whereas lighting professionals have more relativist views.



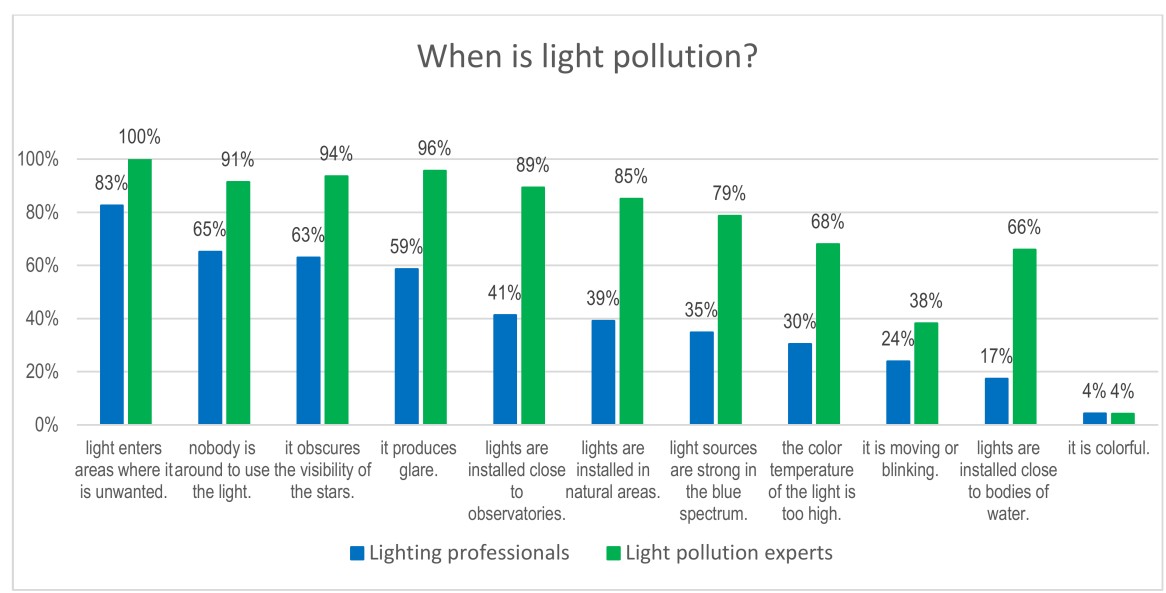

**Figure 2.** Problem dimensions: "In which situations do you consider lighting as pollution?" Percentage of positive answers by group (multiple choice, filter question following the definition "it depends", n = 91, sorted by lighting professionals' feedback).

### 4.2. Why is it a Problem?

The light pollution experts' more acute perception of the problem is also reflected in their evaluation of its different dimensions (Figure 3). When asked why light pollution should be reduced, 82% of the respondents in this group rated *all six* suggested arguments as "important" or "very important", in contrast to 58% of lighting professionals.

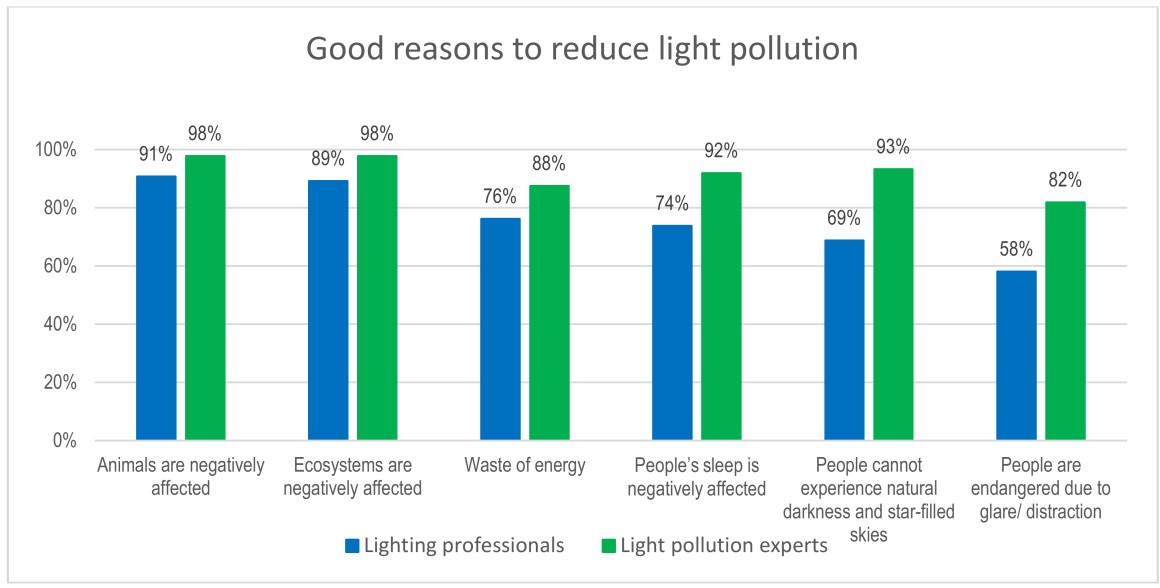

**Figure 3.** Problem dimension: "In your opinion, why should light pollution be reduced? Please indicate how important you find the following..." Percentage of respondents in each group who answered 4 or 5 on a scale from 1-*not at all important* to 5-*very important*. The option "I doubt this is an issue" (−1) was not chosen (sorted by lighting professionals' feedback). See Table S3 for more detail.

Over 90% of all respondents agreed that the *negative effects of lighting on animals and ecosystems* are important or very important reasons to tackle the problem. The least supported argument in both groups—*people are endangered due to glare/distraction*—was still considered important by 82% of the

light pollution experts and 58% of the lighting professionals. This last place on the list may reflect the expert debate on whether glare should be discussed as a form of light pollution. Interestingly, this valuation stands in contrast to the fairly high ratings for glare as a form of light pollution (Figure 3).

When exploring inter-group discrepancies, we see that the inter-group differences are greatest and that the impact of occupation is thereby also statistically significant in regressions with various explaining variables (see footnote 3) for the three last-ranking arguments that regard negative effects on *people's sleep* ($p < 0.05$), *people's incapacity to experience natural darkness and star-filled skies* ($p < 0.01$) and *dangers due to glare/distraction* ($p < 0.05$).

### 4.3. What is the Governance Challenge?

The governance challenge was operationalized in terms of potential obstacles to light pollution mitigation. Again, light pollution experts express stronger opinions in almost all points (Figure 4). The only possible obstacle that does not fit this pattern concerns the *definition of light pollution*. While 50% of the lighting professionals think that the lack of a clear-cut definition is an important impediment, this view is only shared by 32% of the light pollution experts, which corresponds with their more definite understanding of the problem as outlined above.

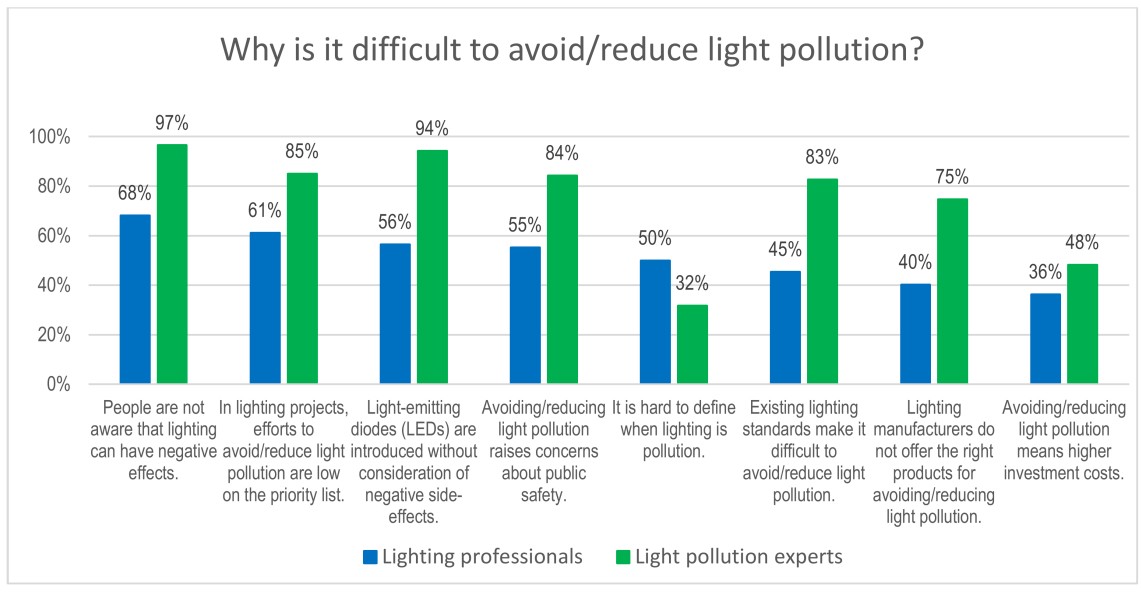

**Figure 4.** Obstacles to light pollution mitigation: "Based on your experience, how relevant are the following potential obstacles to avoiding/reducing light pollution?" Percentage of respondents in each group who answered 4 or 5 on a scale from 1-*not at all important* to 5-*very important* (sorted by lighting professionals' ranking). See Table S4 for more detail.

Focusing on the ranking of the listed items, the top three obstacles for the lighting professionals are first, the *general lack of awareness that lighting can have negative effects* (68%), second, *the low priority of the issue in lighting projects* (61%), and third, *the installation of LEDs without consideration of side-effects* (57%). These potential obstacles are also considered as *most important* by light pollution experts, but they rank the LED problem second (94%) and the low priority of light pollution in lighting projects third (85%). In both groups, the complex issue of *public safety concerns* ranks fourth, albeit very close to the third-most important item.

Looking more closely at the discrepancies between the groups, we find an interesting pattern. The inter-group differences are greatest and statistically significant ($p < 0.001$) in regressions asking for the impact of occupation and for the three items that concern lighting practices (introduction of LED lighting, lighting standards, lack of adequate lighting technology). Light pollution experts consider these potential obstacles as considerably greater than lighting professionals (34 percentage points and

more). Scepticism or even frustration regarding current lighting practices also dominate about half of the open statements to this question: "Mindless installation of harsh, eye-gouging LEDs has become an epidemic worldwide and it just keeps getting worse and worse," writes one light pollution expert. Another criticizes "manufacturers and National agencies ignoring/minimizing light pollution as side effect . . . "

Light pollution experts' negative or sceptical views on current lighting practices could be interpreted as a result of their vigilant observation of, but limited access to, the lighting field. It also supports our assumption that lighting professionals are less critical of their field as they think they could solve the problem if they were invited to give their expertise. Accordingly, one lighting professional proposes to "promote good lighting design" as a way of light pollution mitigation. Another complains that "specification and installation decisions are made by parties without appropriate training/expertise."

The gap between the groups is smaller regarding what can be summarized as *light user-related* obstacles (lack of awareness regarding the negative effects of lighting, low priority of the issue in lighting projects, and concerns about public safety), which are rated amongst the most important impediments to light pollution mitigation. While differences between the groups still range between 24–29 percentage points when considering only the high values (answers 4 and 5), they are even smaller when looking at *all* responses (entire scale from 1 to 5): the difference between the group-specific mean values is below 0.9 with relatively high average scores between 3.6 and 4.8 (see Table S4). The user-related obstacles' importance is also reflected in the open responses (n = 36 for both groups), where almost half of the statements address unawareness, ignorance or misconceptions among light users, including municipalities. "There is a general ignorance and apathy on the issue", remarks a light pollution expert and adds that "slowly but surely people are waking up." A lighting professional criticizes "the poor knowledge and the deficient light culture of politics and city administrations."

### 4.4. Who is Responsible for Tackling the Issue?

Since environmental issues like light pollution raise questions of accountability, we asked the survey participants to attribute responsibility to a list of stakeholders. Again, the responses show that light pollution experts attribute generally more responsibility to each listed group than lighting professionals, but the inter-group differences are smaller than the potential obstacles (Figure 5). In particular, there is broad agreement that lighting designers/planners, politicians and public administration are responsible, followed by lighting manufacturers. The inter-group comparison shows that the lighting professionals in our sample are, as expected, willing to take on responsibility, which is also assigned to them by light pollution experts. The lighting professionals' willingness to tackle the problem is underlined by the responses of the nine respondents who sell lighting products as their main occupation: They all find that lighting manufacturers are responsible. Furthermore, lighting professionals hold the actors in their field more responsible than politicians and public administrations, whereas light pollution experts see them as being roughly equally responsible.

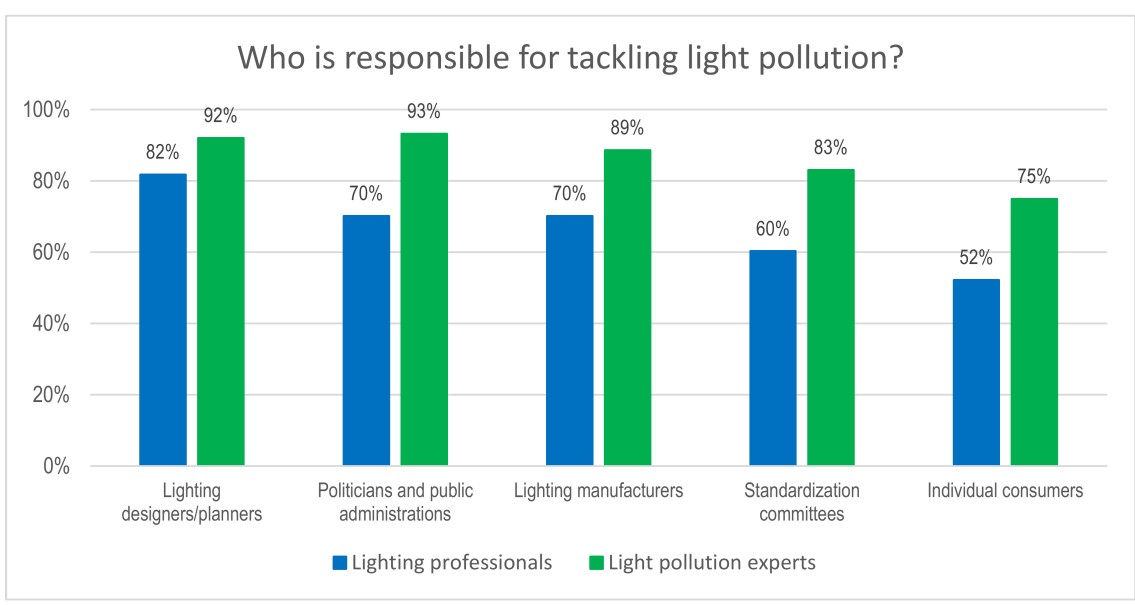

**Figure 5.** Responsibility: "In your opinion, to which degree are the following actor groups responsible for avoiding or reducing light pollution?" Percentages of respondents in both groups who answered 4 or 5 on a scale from 1-*not at all responsible* to 5-*very much responsible* (sorted by lighting professionals' feedback). See Table S5 for more detail.

The trust in professional expertise is also reflected in the responses regarding the responsibility of individual consumers. Only 52% of the lighting professionals in our sample consider them highly responsible (values 4 or 5), whereas 27% think that end users of light are *not* responsible (values 1 or 2). Light pollution experts, too, rank individual consumers least responsible, but hold them considerably more accountable: Only 7% find individual consumers have no responsibility, while 75% think they do. The discrepancy might result from lighting professionals' expert attitudes towards their clients as expressed in some open statements (e.g., "Clients in general ask for more"), whereas light pollution experts might identify themselves as individual consumers who take action against light pollution. This interpretation corresponds with a sense of self-responsibility that was expressed in several open survey statements (Question: "In your light-related activity, do you actively take precautions/action to avoid or reduce light pollution?"). Here, light pollution experts describe how they chose specific technology to reduce light pollution in their immediate surroundings or participate in public campaigns and education to raise awareness for the issue.

Finally, that most light pollution experts (75% and more) rank all items listed in the question highly likely reflects their interest in mobilizing against light pollution and addressing the issue broadly. This is also expressed in open statements regarding responsibilities: "We are all responsible", argues one light pollution expert. Others attribute responsibility to "civil society/local communities (people that live in areas which suffer from too less or too much light)" or to "people in general", as "they are the final users of the lighting systems and have to have a capital role in this issue . . . "

Despite differences in degree, we see that over 70% of the survey respondents in both groups can agree that the main responsibility for tackling light pollution lies with decision makers and institutional actors in the lighting field and in politics. That they hold individual consumers less responsible can be interpreted as a sign of their system understanding of the challenge and sense of realism. After all, most respondents also indicated that the general lack of awareness among light users is a major obstacle to light pollution mitigation, making them a difficult stakeholder group to start with.

*4.5. What Should be Done?*

Regarding possible light pollution mitigation measures, the views of lighting professionals and light pollution experts converge far more than in previous questions, with especially small inter-group differences for high-ranking potential measures. The *promotion of best practice lighting projects,* e.g., *in municipalities,* is unequivocally strongly recommended, i.e., by 95% of all respondents (Figure 6). Moreover, 89.6% of the lighting professionals and 84.5% of the light pollution experts recommend to *use lighting concepts and integrated light planning* to tackle the problem. The great for such strategic policy instruments is remarkable as they are still in a phase of development and not yet very well established in urban and regional light planning practice [50,51]. This might explain why light pollution experts find education measures and awareness-raising even more desirable (mean values between 3.6 and 4.7, see Table S6).

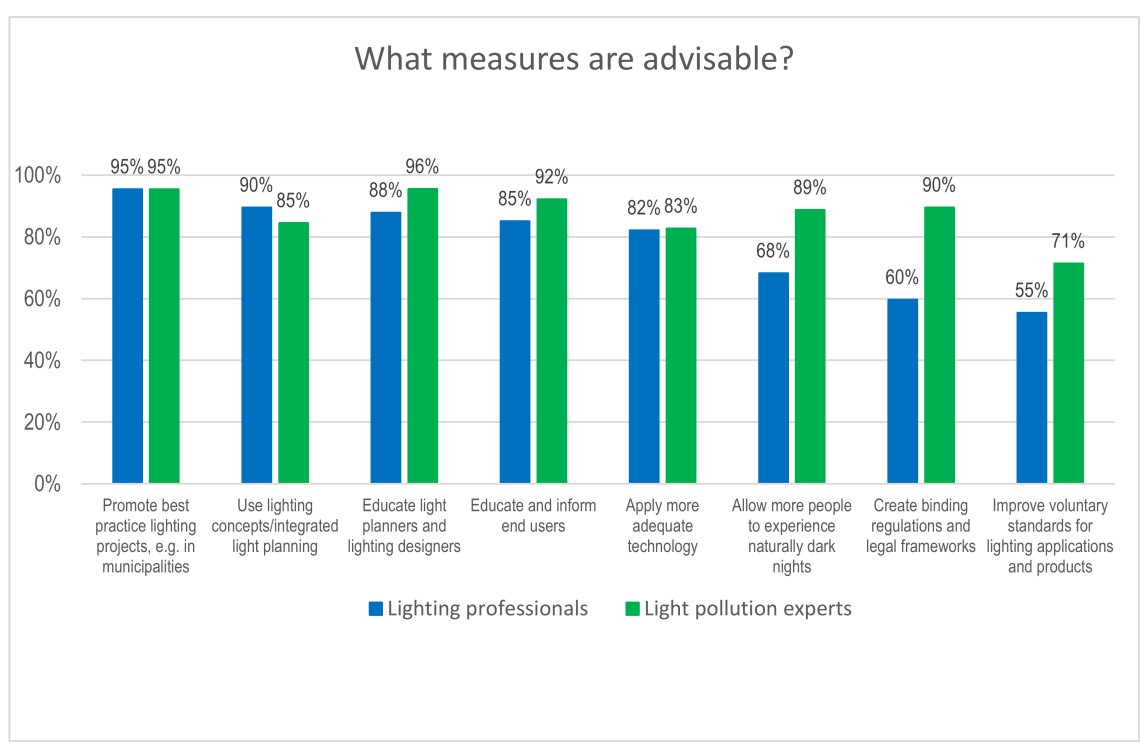

**Figure 6.** Recommendations: "To which extent would you recommend the following measures to avoid/reduce light pollution?" Percentage of respondents in each group who answered 4 or 5 on a scale from 1-*not at all* to 5-*very strongly* (sorted by lighting professionals' feedback). See Table S6 for more detail.

The impacts of occupational backgrounds (lighting professional or light pollution expert) were only statistically significant for the three least important possible recommendations: Light pollution experts more strongly recommend *allowing people to experience naturally dark nights* ($p < 0.01$). They are also significantly more in favor of *mandatory regulations and legal frameworks* ($p < 0.001$) and more strongly recommend *improving voluntary standards for lighting applications and products* ($p < 0.05$). The discrepancy, especially with regard to binding regulations, partly matches our assumption that lighting professionals would rather support the better use of existing expertise and technology. Yet, we also see that more than half of them recommend the creation of regulations and better standards for tackling light pollution. Meanwhile, the share of light pollution experts who recommend better regulatory frameworks is smaller than we expected. This slightly diminished enthusiasm might result from the experience that rules and regulations need to be understood, followed and enforced if they are to make a difference, which has proven problematic in the lighting field [16] (pp. 151–152).

Seen overall, it is relevant that the respondents show a high degree of agreement to *all* suggested measures (agreement above 50% in both groups for all items). One open statement (Question: Why do you find specific measures more advisable than others?) explains how the items complement each other: "We need a mix of (1) measures to avoid mistakes (education; binding regulations); (2) retrofitting programmes to fix past mistakes; and (3) more dark sky parks experience to spread the benefits of low light pollution." Another points out that outdoor lighting has historically been designed and installed with little regard for its potential negative impact: "Overcoming this will take a coordinated approach using awareness raising, education, rules and regulation cross the range of the industry including end users."

## 5. Discussion: Differences, but Common Grounds

The findings of our expert survey both confirm and challenge our group-specific expectations (Section 3). While it is important to keep in mind that the lighting professionals in our sample are likely particularly sensitive to and reflexive about light pollution issues due to a self-selection bias, the analysis of the survey results nonetheless point towards both shared and conflicting views.

### 5.1. Problem Perception: Absolute and Situated Definitions across Expert Groups

First of all, the results highlight that light pollution experts and lighting professionals can agree on relevant points. This is most evident in the unexpected result that most lighting professionals in our sample accept the concept of light as pollution (Figure 1). More than half of all respondents perceive light pollution as a situation-dependent phenomenon. The four top-ranking critical situations in both groups (Figure 2) describe light trespass, unused light, sky glow and glare. Focusing on disagreement, it seems that the most important difference is that light pollution experts define light pollution in more absolute terms, for instance when light shines in natural areas, irrespective of its specific purpose or use. This discrepancy seems important when it comes to possible counter-measures, as an essentialist understanding calls for zoning and thresholds in regulatory approaches, whereas a relativist understanding calls for deliberation and the negotiation of conflicting interests.

Regarding the potential dimensions of the problem (Figure 3), we found support for our assumptions that light pollution experts overwhelmingly consider the lost experience of natural darkness and the night sky as a relevant reason for reducing light pollution, whereas lighting professionals find other aspects such as energy savings more important. Interestingly, they rated aspects related to the human experience of light and darkness least important. This was unexpected in light of professional debates on visual comfort and human-centric lighting.

Given that ecology is rarely an issue in lighting projects, the almost unanimous perception of unwanted side effects on flora and fauna as being highly relevant was surprising. This can be understood as a sign that the lighting field is ready to take into account the increasing scientific evidence on effects of ALAN. Recent publications by lighting designers support this conclusion [52], calling for "biologically benign forms of energy-efficient lighting" and transdisciplinary efforts by physicists, engineers, medical experts, biologists, designers, planners, regulators and policymakers to "minimize the negative impacts of artificial lighting at night, indoors and out." [43] (p. 274).

We conclude that a shared concern for the unwanted environmental effects of ALAN could constitute common grounds for lighting professionals and light pollution experts. After all, tackling light pollution is also perfectly in line with energy-saving goals as a positive side effect and another selling argument for light pollution mitigation.

### 5.2. Governance Challenge: Raising Awareness is Key

Regarding challenges and opportunities for tackling light pollution, we see a broad consensus that there is a problematic and general unawareness of the problem (Figure 4): More than two-thirds of the respondents agree that light pollution mitigation is hampered by the fact that people are not aware that light can have negative effects. This result fully confirms our expectations regarding light

pollution experts. For lighting professionals, it indirectly supports our assumption that they consider themselves unsolicited experts, especially as they point toward the low priority of light pollution in lighting projects as the second most important obstacle. In other words, lighting professionals seem convinced that the problem could be managed if they were only asked more often to offer their expertise. Yet, this implies that clients are aware of the problem in the first place.

Moreover, both groups express their concern that the introduction of LED lighting endangers light pollution mitigation. As expected, these responses reflect the light pollution experts' concern that the worldwide increase in blue-rich LED lighting will intensify the problem, and lighting professionals' dissatisfaction with bad or badly installed LED technology. It is important to note, however, that light pollution experts significantly more often see LED lighting as an important impediment to light pollution mitigation. The same applies to the lighting industry and existing lighting standards, which light pollution experts consider much more problematic than lighting professionals.

Taken together, the group-specific responses to our question regarding potential obstacles differ more than those regarding the problem and possible solutions. We interpret this as a result of different practical experiences: While lighting professionals are perfectly familiar with, and therefore criticize, imperfect project realities, light pollution experts observe the lighting field from their concerned outsider perspectives as astronomers, environmentalists, researchers or citizens and criticize the entire system.

### 5.3. Possible Solutions: Who will Tackle the Problem of Light Pollution and How?

Finally, expert opinions on possible solutions have great practical relevance. Here, our assumptions regarding responsibility attribution were only partly confirmed. Both groups hold individual consumers least responsible and lighting designers and planners most responsible, followed by politicians and public administrations (Figure 5). As expected, the lighting professionals in our sample are overwhelmingly ready to take on the challenge. Meanwhile, light pollution experts unexpectedly attributed responsibility to practically anyone, including themselves ("all of us"). Thus, it seems that contrary to many environmental debates where scapegoating prevents action [53], the majority of the survey respondents appeared to be ready to take action. This is also reflected in their answers to another open question, where we asked whether they "actively take precautions/action to avoid or reduce light pollution": The numerous responses showed a broad range of activities, including the private use of adaptive lighting systems, the promotion of darkness in municipal lighting schemes and public dark-sky initiatives [16] (pp. 199–202).

The respondents' recommendations on how to tackle the problem are surprisingly consensual across the two groups (Figure 6). The promotion of best practice in lighting projects is highly recommend by 95% of the survey participants. They also broadly agree that integrated light planning is a promising measure. This common ground seems to be particularly relevant, as these instruments are not yet widely established, but municipalities seem increasingly open to rethinking their lighting schemes in response to the profound technological transition, climate change policies and an increasing public concern for light pollution [50]. Moreover, lighting professionals have begun to explicitly consider light pollution concerns when they develop lighting schemes and guidelines [54,55]. These planning-oriented recommendations send a clear message to policy makers, as they can inform best practice, make the philosophy of demonstration projects more tangible and offer guidance to decision makers and light planners.

In line with the widely-shared view that lacking awareness of light as pollution is a key obstacle to tackling the problem, education of both lighting experts and users is a highly consensual recommendation. In comparison, regulatory measures like binding law were less consensual. We thus conclude that the largest common grounds among the two groups exist regarding soft measures, such as setting good examples and raising awareness. From a policy perspective, such educational and best-practice measures constitute low-hanging fruit, as they are likely to raise attention and enthusiasm with less controversy and opposition than may be the case with hard regulation. The experts who

participated in our survey seem well-aware of this across the board. However, regarding the question of whether such soft measures will be enough, our survey indicates that opinions differ.

## 6. Conclusions

This exploratory study focuses on expert views on light pollution to explore common grounds for future debates and political strategies. Our findings show that lighting professionals who provide illumination and light pollution experts, who problematize artificial light at night, do not necessarily live and work in worlds apart. Instead, the lighting designers, planners and manufacturers that participated in our survey are ready to take into account the increasing scientific evidence on negative non-visual effects of artificial light at night on ecosystems in order to provide better lighting as part of their business. They accept the notion of light pollution, especially in situations where lighting does not live up to quality standards of light engineering, planning or design. Finally, they largely agree with astronomers, environmentalists, researchers and dark-sky activists that there is a need for raising awareness for the unwanted side-effects of lighting.

While light pollution is far from being a mainstream topic, these findings suggest that there is a rising awareness for the problem and its potential effects not only in science and society, but also in the lighting field. As an emerging environmental concern, the issue also raises questions that go beyond this study. For instance, the problem and solutions seem to be less controversial than the obstacles that prevent light pollution mitigation. To better understand the governance challenge, it therefore seems advisable to not only study the effects of ALAN, but also the societal and cultural contexts in which it is produced, used and changed [13,25,47]. Disagreement and controversies can thereby offer a salient starting point for understanding the values and fears, path dependencies and future visions associated with light and darkness [19,39,56].

Moreover, the governance challenge of light pollution shows interesting parallels with other environmental issues that are more established and may offer instructive insights, like noise or chemical pollution [57]. One lesson learnt is that patchy knowledge bases and scientific uncertainty constitute a challenge for risk communication [58]. With regard to our results, this raises the sensitive question of how to create public awareness for light pollution without either dramatizing or downplaying its potential effects. Research alone cannot meet this challenge. Instead, it highlights the need for science-policy interfaces as well as inter- and transdisciplinary exchange. A number of initiatives and organizations in both the lighting field and the emerging ALAN research community offer platforms to meet this demand [16] (pp. 224–246). However, although experts of both fields have started to exchange views and knowledge, a joint platform for opening and closing debates is still missing, which can cause uncertainty among light users [25].

The commonalities between two fields of expertise highlighted in this paper thus have practical relevance, as they can facilitate exchange between experts that share an interest in light pollution mitigation despite their diverse, and potentially opposing, professional backgrounds. Fostering this exchange seems pivotal since light pollution mitigation means a transformation of lighting practices and the positive connotation of lighting [13]. It is even more important as the strong support for situation-specific definitions of light pollution in our study suggests that the issue can hardly be solved only in principle, but calls for negotiation at the level of lighting projects and in public discourses. Seen from this perspective, the recommendation to raise awareness and education seems to be a 'no-brainer'. Instead, it raises the question about what should be taught to whom and how. In this respect, critical debates on light planning and lighting design that engage local residents and stakeholders from lighting and non-lighting backgrounds on-site in concrete projects seems a promising starting point. After all, there is no shortage of technological options or visionary concepts like "human centric" or "smart" lighting, but a lack of projects that realize these possibilities in sustainable ways. This is not surprising, as determining what "sustainable" lighting actually is, will very likely stir debate, opposition and controversies over means and goals. It is also quite possible that lighting professionals and light pollution experts who agree in principle will disagree when it comes to concrete decisions.

Therefore, public testing and reality checks of visionary ideas and concepts are essential. They bear potential for improvement, mutual understanding and, most importantly, bring the global discussion on light pollution into a local, practical context, and make it real in its consequences.

**Supplementary Materials:** The following are available online at http://www.mdpi.com/2071-1050/11/6/1696/s1, Table S1: Variables used in regressions, Figure S1: Regressional analyses, Table S2: Trends, Figure S2: Trends, Table S3: Problem dimension, Table S4: Obstacles, Table S5: Responsibilities, Table S6: Recommendations. An overview of variables is available at www.ufz.de/light-pollution.

**Author Contributions:** N.S.-R., J.M. and E.D. have conceived and conceptualized the study and questionnaire. N.S.-R. and M.S. prepared and carried out the survey and analyzed the survey data. N.S.-R. and J.M. wrote the first draft of the paper and all four authors contributed to writing, revising and editing the manuscript.

**Funding:** This research received no external funding.

**Conflicts of Interest:** The authors declare no conflict of interest.

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
