# Peer review of "Lighting Professionals versus Light Pollution Experts? Investigating Views on an Emerging Environmental Concern"

_sustainability, doi:10.3390/su11061696_

Reviewer 1 Report

Dear authors, thank you for this research! I should note that I would fall into the "light pollution" expertise although I have dappled with industry and lighting design experts. I find this work incredibly important and timely. Here, you used a survey of both light pollution experts and light design experts to see how divided the two camps are and you show that the divide is not as big as one would think. This is very hopeful. Of course, you show that lighting designers are less harsh about light pollution overall whereas light pollution experts are more dogmatic. This is very important to show and now that we know the middle ground, we can begin to work together as light pollution experts and light design experts to reduce unneeded light in the environment, which will benefit everyone (other than maybe light bulb companies).

This research is very important as it shows people from both camps what the other camp's point of view is. I find your manuscript well written and the methods sound. Other than a few typographical changes, I have only one major suggestion and that is I would like for your last paragraph of your discussion to talk more about bridging the gaps between these two camps. This of course will be your opinion, but how do you think we could get these two camps to be on the same page and work together more? What can be done? You end with "it raises the question of what should be taught to whom and how." I want you to propose a couple of answers to your own question.

Minor comments:

Table 1: "mail" to "male"

              last statement bottom right - "an interest in"

168: we refer to water organisms as "aquatic"

168-179: A very long run-on sentence here, it was difficult to get through. I would suggest turning this into more than one sentence.

414: Write out 27 as it starts the sentence.

Author Response

Comments and Suggestions for Authors

Dear authors, thank you for this research! I should note that I would fall into the "light pollution" expertise although I have dappled with industry and lighting design experts. I find this work incredibly important and timely. Here, you used a survey of both light pollution experts and light design experts to see how divided the two camps are and you show that the divide is not as big as one would think. This is very hopeful. Of course, you show that lighting designers are less harsh about light pollution overall whereas light pollution experts are more dogmatic. This is very important to show and now that we know the middle ground, we can begin to work together as light pollution experts and light design experts to reduce unneeded light in the environment, which will benefit everyone (other than maybe light bulb companies). This research is very important as it shows people from both camps what the other camp's point of view is. I find your manuscript well written and the methods sound. Other than a few typographical changes, I have only one major suggestion and that is I would like for your last paragraph of your discussion to talk more about bridging the gaps between these two camps. This of course will be your opinion, but how do you think we could get these two camps to be on the same page and work together more? What can be done? You end with "it raises the question of what should be taught to whom and how." I want you to propose a couple of answers to your own question. 

Minor comments:

·      Table 1: "mail" to "male", last statement bottom right - "an interest in"

·      168: we refer to water organisms as "aquatic" (now 175)

·      168-179: A very long run-on sentence here, it was difficult to get through. I would suggest turning this into more than one sentence. > We supposed you meant the sentence that is now starting in line 180 (before 179)? Thanks, we changed it which made the content even clearer.

·      414: Write out 27 as it starts the sentence. (we have changed the sentence, also the one on p. 3, line 112, which started with “205 participants…”)

Thank you very much for your positive and encouraging feedback and for reading our paper so closely. We had it spellchecked.

We also very much liked your suggestions for our conclusion and added a paragraph (p.17-18,  618-628).

The revised manuscript is attached.

Reviewer 2 Report

Well done! I found it very interesting for anybody tackling the light pollution topics, either form research, light-pollution activist or lighting design side and I highly recomend to publish it. It contains a lot of interesting facts and conclusions, and yes, in the arena of light pollution we humas, as entites, do play a large role, and our behaviour and ways of thinking do infulence our actions about the light pollution problems, making such articles highly valuable.

Author Response

Thank you very much for your fundamental support of our paper and your positive feedback. This is very encouraging!

Reviewer 3 Report

First of all it is important to say this paper propose an original comparison study of the different approaches to a pollution problem from different points of view.

But in my opinion there are two big problems.

- Number of participants used in the survey. Finally just around 100 it seems poor in comparison with the expected population. But of course this is not easy to solve

- Statistics. Now there is almost no description or any clarification of data analysis. This has to be included and discussed to evaluate the confidence of the results.

Finally just a short comment, there is a citation to footnote 3 and I have not find it.

Author Response

Thank you very much for your overall positive feedback and your helpful suggestions. Regarding your critique of our research design, we see your point but think it does not apply to our study. In other words, while of course a larger population would have been most welcome, we do not think that the number of participants is a problem for this paper as we clearly state that the study is exploratory. We have added a sentence in our methods section to make it clearer (p.3, 100-102).

This might be unusual as many surveys are designed to be representative and test hypothesis, but a mixed-methods approach such as ours is wrong or illegitimate, as long as the limitations of this approach are made transparent. Accordingly, we do not “test hypothesis” but refer to and discuss assumptions. We also do not claim to present representative data and results, but call our study exploratory. The aim was not to characterize the two groups in an essentialist way (this is how they are), but to challenge and explore the idea of “two camps”. This revealed signs of inter-group convergence, which we substantiate in the study with qualitative and quantitative data.

- Statistics. Now there is almost no description or any clarification of data analysis. This has to be included and discussed to evaluate the confidence of the results.

Again, this is not a representative study and the regressions only play a subordinate role. However, we have clarified our approach on p.3, footnote 3, p.3, 128, 353, …). If necessary, we can provide more information than in footnote 3 (the one you could not find) as part of the supplementary material or on our website (www.ufz.de/light-pollution.de, see also p.17, line 628-631).

Finally just a short comment, there is a citation to footnote 3 and I have not find it.

This is the footnote on p.3 that explains what we did in the regressions. We don’t know why you could not see it. It might be a problem with Word Windows 10. One of the co-authors also could not see footnote 2 and 3. We could insert the information in the main text. We leave that up to the editor(s) since we prefer the footnote.

The revised manuscript is attached.

Reviewer 4 Report

Dear authors,

very interesting paper and extremely need it.

I find only a few details that I find that need to be improved. First, need to be explained in which languages the form was prepared. If it was only English, that can be an important bias that should be mentioned. Also, it would be interesting to know how significant is the sample on each population, for example, comparing the size of the sample with the number of attendees of the most important congress for lighting professionals and light pollution experts. 

My experience is that the light pollution experts are much better represented that the light professionals.  Although, I think that the results are very important, at is showing the peak of the iceberg.

Also, the authors should mention that some of the answers were already given, so might be other potential solutions not mentioned, like create a light installation controls like exist in some  

regions of Spain, that was mentioned by Sánchez de Miguel 2015 (Annex) https://zenodo.org/record/1289933#.XH3B6Yj0nFg

as one of the critical factors to make a light pollution control regulation to be effective.

Author Response

English bias is a very good and important point. We have added a comment on this (p.3, 95-.97). We have tried to add information on “how significant” our small and biased sample is (based on twitter followers and PLD-C attendees), but found it too vague. However, we have made it more explicit that our sample is not representative (p.3, 100-102).

My experience is that the light pollution experts are much better represented that the light professionals.  Although, I think that the results are very important, at is showing the peak of the iceberg.

We agree, it’s only the peak of the iceberg, especially as our paper only captures the views of lighting professionals that are already aware of the issue. Regarding the representation: This depends on the focus. Lighting professionals are very well represented when it comes to established topics and issues like light-technical standards. Their institutions are over 100 years (illuminating engineering societies) or are gaining importance (lighting design associations like IALD).

Also, the authors should mention that some of the answers were already given, so might be other potential solutions not mentioned, like create a light installation controls like exist in some  regions of Spain, that was mentioned by Sánchez de Miguel 2015 (Annex)  https://zenodo.org/record/1289933#.XH3B6Yj0nFg as one of the critical factors to make a light pollution control regulation to be effective.

Of course, there are more aspects than could be covered in multiple-choice questions of the questionnaire. We added a comment on this as well as the suggested information (194-196) including the reference to Alejandro Sanchez de Miguel’s Dissertation.)

Thank you very much for your positive feedback and helpful suggestions!

The revised manuscript is attached.

Round  2

Reviewer 3 Report

I suggest to be as clear as possible with the methods used so the use of Supplementary Materials can be a good idea.

Good paper in any case.

Author Response

Dear reviewer, once again, thank you very much for having a second look on our article!

We very much appreciate your feedback and have now upload supplementary material that shows as clear as possible what we have done (Table S6 and Figures S2).    

Best regards and thanks again,

the team of authors